# Removal of Dye from Aqueous Solution Using Ectodermis of Prickly Pear Fruits-Based Bioadsorbent

**Fatimah A. M. Al-Zahrani** [1,*] **, Badria M. Al-Shehri** [1,2,3] **and Reda M. El-Shishtawy** [4,5,*]

1  Chemistry Department, Faculty of Science, King Khalid University, P.O. Box 9004, Abha 61413, Saudi Arabia
2  Research Center for Advanced Materials Science (RCAMS), King Khalid University, P.O. Box 9004, Abha 61413, Saudi Arabia
3  Unit of Bee Research and Honey Production, Faculty of Science, King Khalid University, P.O. Box 9004, Abha 61413, Saudi Arabia
4  Chemistry Department, Faculty of Science, King Abdulaziz University, P.O. Box 80203, Jeddah 21589, Saudi Arabia
5  National Research Centre, Dyeing, Printing and Textile Auxiliaries Department, Institute of Textile Research and Technology, 33 EL Buhouth St., Dokki, Giza 12622, Egypt
*  Correspondence: falzhrani@kku.edu.sa (F.A.M.A.-Z.); relshishtawy@kau.edu.sa (R.M.E.-S.); Tel.: +966-503813092 (F.A.M.A.-Z.)

**Abstract:** Billions of grams of ectodermic fruits, such as prickly pear fruits, are removed and thrown as waste worldwide. In this study, an inexpensive approach was used to successfully transform the agricultural waste prickly pear fruit peels (PPFP) into a new adsorbent used to remove dye (PTZIDM). The adsorbent PPFP revealed a porous structure and a fair surface area. It was characterized and evaluated using a scanning electron microscope (SEM), Fourier transform infrared spectroscopy (FTIR), and surface area measurements (BET). The effectiveness of the PPFP's adsorption was assessed in relation to pH, PPFP dose, contact time, and initial dye concentration. The kinetics and isotherm characteristics were investigated. More than 95% removal efficiency was obtained within 60 min at the adsorbent dose of 0.1 g for an initial dye concentration of $1 \times 10^{-5}$ M at pH 3. The pseudo-second-order models and the Langmuir isotherm are excellent at explaining the characteristic of dye adsorption. This work offers a rapid and simple method for efficiently converting biomass waste and using it to remove pollutants.

**Keywords:** adsorption; water-insoluble dye; prickly pear fruits; kinetics; isotherm analysis; agricultural waste

## 1. Introduction

The two ecological problems of the twenty-first century that have the greatest impact on the environment are water pollution and environmental change. Water pollution is quite concerning because the effluents from the textile sector include so many synthetic colors, many of which are dangerous contaminants [1]. Therefore, it is essential to control the amount of dye in wastewater before discharging it into the environment. Since most synthetic dyes are not biodegradable and have a high level of oxidation resistance, removal can be challenging. To cope with colors, organic debris, heavy metals, and other impurities, removal techniques encompass photocatalysis, membrane treatment, adsorption, and advanced oxidation processes.

Adsorption is commonly employed to eliminate contaminants, including heavy metals and dyes since it is a secure, efficient, clean, and technically possible procedure. Because of this, the adsorption method is a fascinating separation method that uses an appropriate substance known as an absorbent. This substance has a relatively greater surface part and absorbency, enabling a reasonable adsorption symmetry kinetic [2,3]. This technology is the widely utilized technique of removing water pollution due to several factors, including

its high efficiency, eco-friendliness, low price, ability to eliminate both organic and inorganic pollutants, penetrable and porous substantial resistance to toxins, high efficiency, unsophisticated design, and ease of operation [4–6]. Pollutants are transferred from an aqueous solution to an adsorbent during the adsorption process [7]. To establish an adsorbent's appropriateness and applicability in pollutant removal, equilibrium models, kinetics, and thermodynamic parameters should be examined using experimental data [8,9]. The Langmuir, Temkin, Freundlich, Halsey, and Dubinin–Radushkevik isotherm patterns were utilized to investigate the adsorption data. On the other hand, models for intra-particle diffusion and pseudo first and second orders have been frequently used to characterize kinetics data [10–14].

It is essential to mention that husks, fruit shells, peel, and seeds are particularly beneficial as adsorbents since employing these in wastewater handling boosts their usefulness while easing the discarding issues [15–17]. Moreover, it is reported that activated carbon is frequently used to treat wastewater to eliminate pollutants. This is typically made from carbon-rich substances such as wood, lignite, coal, cooking oil, etc. Several teams are working to create suitable adsorbents from these types of spoils and waste [18,19], and stimulated carbon is typically a significant adsorbent in eliminating contaminants and impurities from wastewater due to its larger precise surface area and the supplementary active places. Activation has the drawback of being expensive and using a lot of energy. Therefore, it is crucial to find novel, inexpensive materials that may be used as adsorbents [20].

Prickly pear fruits are a sort of the genus Cactus in the family Cactuaceae. Its fruit, cladodes, and flowers have historically been utilized. Because of their nutritional properties, they are used in folk medicine [21]. According to various reports, the peel of this fruit accounts for approximately 50% as a percentage of its total weight [22]. As a result, every year, massive amounts of kilos of fruit peels are wasted, resulting in landfilling and ecological disasters, whereas proper use of these waste materials may result in a significant new therapeutic resource. The fruits of the prickly pear have a high medicinal significance, with anti-ulcerogenic [23], neuroprotective [24], antioxidant [25], anticancer [26], and hepatoprotective properties [27]. Prickly pear fruit juice contains polyphenols. It is high in flavonoids and ascorbate, making it a good supplier of vitamin C [28].

Some prickly pear preparations (fruit peel, mucilage, and cladodes) were tested as bioadsorbents for dye removal. Prickly pears were subjected to different treatment methods, such as simple heat and/or chemicals, prior to use. Cladode-activated carbon form (with $H_3PO_4$ at 450 °C) [29] has been utilized to eliminate methylene blue and iodine. The fruit peels showed a great opportunity as a biosorption for methylene blue [30]. The capacity of adsorption was 222 mg/L when the fruit peels were boiled in water, solar-dried for 20 days, cleaned with distilled water, dried (40–50 °C), and smashed (0.315 mm). Similarly, dried peels treated with sulfuric acid (1 M) and sodium chlorate yielded an effective bioadsorbent at 20 °C and pH 3, the maximum adsorption capacity of brilliant green was 167 mg g$^{-1}$ [31]. Sodium hydrate and sodium chlorate were also used to treat other fruit peel extracts separately. Sodium perchlorate and sodium hydroxide dramatically increased decolorization percentage up to 96% [32].

In this empirical research, peels of prickly pear (*Opuntia ficus indica*), a readily available adsorbent from the Asir area, were tested for its ability to absorb the PTZIDM dye. FTIR and SEM were used to produce and characterize the natural PPFP. On dye adsorption onto PPFP, the impact of several factors involving contact time, adsorbent dose, and pH were assessed and examined. Adsorption isotherms were investigated.

## 2. Materials and Methods

### 2.1. Materials

The chemicals utilized in this study, including NaOH, 10H-phenothiazine, 1-bromohexane, and KI, are reagent grade chemicals, ≥98%, and were purchased from Sigma-Aldrich (Saint Louis, MO, USA). In addition, 10-hexyl-10H-phenothiazine, 10-hexyl-

10H-phenothiazine-3-carbaldehyde, and PTZIDM were synthesized using the previously described methods [33,34].

*2.2. Methods*

2.2.1. Bioadsorbent Preparation and Characterization

Bioadsorbent Preparation

Newly split prickly pear fruit peel (PPFP) was gathered from Alsoda in the Asir region in Saudi Arabia. PPFP was repeatedly purified with distilled water and dehydrated at temperatures ranging from 50 degrees Celsius. The obtained material was crushed in mortar and pestle, passed through 45 micro sieves and treated in an alcoholic solution under a sonicate for half an hour, then cleaned with distilled water and dried with acetone according to Figure 1.

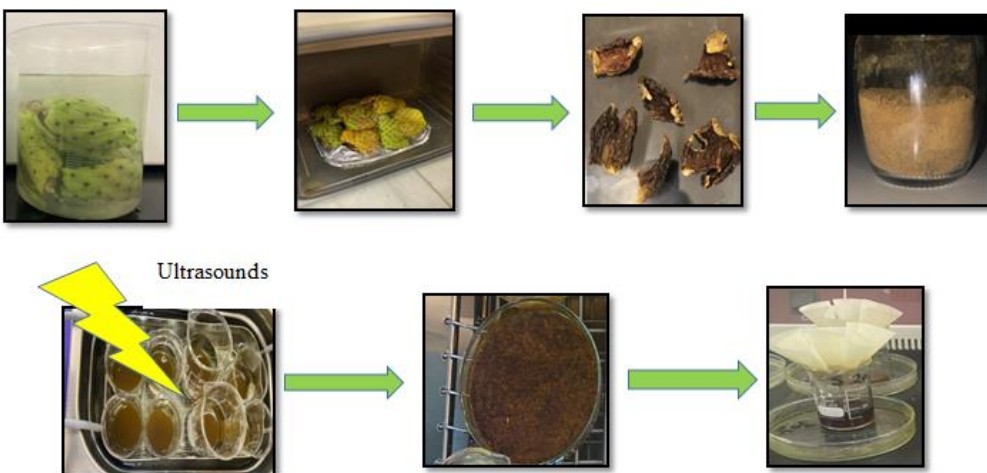

**Figure 1.** A graphical representation for the preparation of the adsorbent.

Characterization

To study the chemical activities of the skin, PPFP was described using the Perkin–Elmer spectrum RXI FTIR spectrometer (PerkinElmer, Beaconsfield, UK). The surface characteristics of PPFP were demonstrated using scanning electron microscope (HIROX SH 400M, Bruker, Bremen, Germany). The specific total surface area was measured by the N2-sorption. isotherms on a QuantaChrome Autosorb-6B (QuantaChrome Instruments, Boynton Beach, FL, USA). The point of zero charge (PZC) was determined using the reported method [35]. The crystalline structure was determined by an X-ray diffractometer (XRD) (Shimadzu Corporation, Kyoto, Japan) [35].

2.2.2. Dye Preparation

PTZIDM was synthesized according to Scheme 1, as previously reported [34]. Thus, phenothiazine was N-alkylated, followed by Vilsmeier formylation to afford the corresponding 10-hexyl-10H-phenothiazine-3-carbaldehyde in good yield. Knoevenagel condensation reaction of the carbaldehyde with 2,2′-(1H-indene-1,3(2H)-diylidene) dimalononitrile afforded the corresponding PTZIDM. The charts of analysis are provided in the Supplementary File.

**Scheme 1.** Synthetic procedure for PTZIDM dye.

*2.3. Adsorption Tests*

The dye (535 mg) was dissolved in one liter of deionized water (water, HPLC grade, Milli-Q apparatus, Sigma-Aldrich) to make stock solutions of PTZID 0.001 M. Using the produced material, the impacts of contact time, adsorbent dose, initial concentration, solution pH, and temperature on the adsorption were examined. All pH readings were measured using a pH meter, and 0.1 M hydrochloric acid and 0.1 M sodium hydroxide were used to change the solution's pH. A UV–Vis spectrometer with a wavelength of 518 nm for PTZIDM dye was used to measure the final dye concentration after the supernatant was collected and filtered at a given time. The quantity of dye adsorption per gram of the prepared PPFP ($q$) at equilibrium ($q_e$) or at a time ($q_t$) was calculated using

$$q_t = \frac{V}{m}(C_o - C_t) \tag{1}$$

$q_t$ (mg/g) denotes the amount of dye adsorbed in each unit weight of the adsorbent; $C_o$ and $C_e$ (mg/L) denote the dye concentration at initial and equilibrium, respectively. $V$ (L) represents the dye solution volume, and $m$ (g) represents the adsorbent weight. The percentage of dye removal was calculated using Equation (2).

$$\% \text{ R} = [(C_o - C_e)/C_o] \times 100 \tag{2}$$

2.3.1. Adsorption Kinetics

Pseudo-first order, pseudo-second order, Elovich, and intra-particle diffusion models were applied significantly to provide information on the adsorption rate of the adsorbent, as shown in Table S1, S means hereinafter Supplementary Materials [36–43]. The kinetic studies may explain the mechanisms and the dye adsorption rate onto adsorbents to examine further and explore the kinetics of adsorption.

Pseudo-First-Order Model

If first-order kinetics is used, the log ($q_e - q_t$) against $t$ in Equation (S1) should yield a linear relationship from which the constants k1 can be calculated.

Pseudo-Second-Order Model

The data on sorption were also defined as a pseudo-second-order method [37]. If second-order kinetics is used, the plot of $t/q_t$ against $t$ in Equation (S2) should yield a linear relationship from which the constants $q_e$ and k$_2$ can be calculated.

Elovich Kinetic Model

The Elovich kinetic model (Equation (S3)) has been used to simulate the gas adsorption process on surface sites and solid systems and has been used to investigate the separation of contaminants from aqueous systems. The Elovich kinetic model aids in the prediction of a system's mass and surface diffusion, activation and deactivation energy. According to the model, the rate of solute adsorption decreases exponentially as the amount of adsorbed solute increases.

Kinetic Model of Intra-Particle Diffusion

Three steps can be used to describe intra-particle diffusion, Equation (S4): (i) The transfer of sorbate from a bulk solution to the sorbent's external layer via molecular diffusion, also called external (or) film diffusion. (ii) Internal diffusion, the movement of sorbate from the surface of a particle into its interior. (iii) Solute particle sorption from active species into the internal surfaces of the pores. The slowest, rate-limiting step will manage the overall percentage of the dye adsorption. The properties of the solute and sorbent can be used to study the type of rate-limiting step in a batch mode. The intra-particle diffusion approach illustrated by Weber and Morris [39] can be used in adsorption, where the diffusion coefficient could be the rate-limiting stage. Weber and Morris' [39] equation calculates the rate constants for intra-particle diffusion ($k_{diff}$). The slope of the straight-line portions of the plot of $q_t$ versus $t0.5$ is $k_{diff}$. These plots have mostly a double nature, i.e., initial curve portions and final linear portions. The initial curved parts are because of surface diffusion effects. Intra-particle diffusion effects cause the final linear segments. When the linear portions of the plots are extrapolated back to the axis, the intercepts (c) are proportional to the thickness of the boundary layer [39].

2.3.2. The Isotherms of Dye Adsorption

The adsorption isotherms indicate the isotherm type of dye adsorption at various starting concentrations at equilibrium. Adsorption isotherms give a complete explanation of the characteristic of the interface between adsorbates and adsorbents. For this kind of study, several equations for isotherms were created and utilized, including the crucial isotherms of Langmuir and Freundlich.

Langmuir Isotherm Model

A little interface between adsorbed molecules was observed with a small number of recognized sites, where the monolayer adsorption on a surface was modeled using Langmuir's isotherm. Furthermore, the monolayer's saturation level impacts the maximal adsorption [40,44]. The model can be seen from the linear equation as depicted in Equations (S5) and (S6) in Table S1. When $C_e/q_e$ is charted against $C_e$, a straight line with a slope of $1/q$ max and an intercept of $1/q_{max}K_L$ is formed.

Freundlich Isotherm Model

According to [41], the Freundlich model calculation supposes that adsorbents have heterogeneous surfaces with various adsorption capacities that each kind of position will adsorb particles as in the Freundlich Equation (S7) in Table S1.

By graphing $\log q_e$ vs. $\log C_e$, which produced an upright line with a gradient of $1/n$ and a divert of $\ln K_F$, it was possible to determine that $K_F$ is persistent (a function of the power of temperature and adsorption) where n signified an invariable linked to the strength of adsorption. The value $(1/n)$ is in the range of 0 and 1, and it represents the degree of surface heterogeneity or adsorption intensity for a favorable process. A number that is closer to zero indicates higher surface heterogeneity [45,46].

Dubinin–Radushkevich Isotherm Model

The Dubinin–Radushkevich model is commonly used to demonstrate the adsorption process onto a heterogeneous surface with a Gaussian energy distribution. The method has

frequently fitted data with high solute activities and intermediate concentration ranges. The strategy was commonly used to distinguish physical and chemical adsorption of dyes with its mean free energy, E per molecule of sorbent (for trying to remove a molecule from its position in the adsorbents space to infinity), which can be calculated using the relationships (S8)–(S10) in Table S1.

Temkin Isotherm Model

The Temkin model proposes linearly decreasing adsorption heat with uniform distribution of the bounding energy rather than logarithmic with coverage. This isotherm model calculates the value of sorption heat (B) by plotting $\ln C_e$ against $q_e$ in Equation (S11) in Table S1. From the values of the slope and the intercept $A_T$, B, and $b_T$ can be calculated.

### 2.3.3. Thermodynamics of Adsorption

In order to achieve the best results, adsorption thermodynamics tests were carried out at different temperature levels (between 25 and 75 °C).

These thermodynamic parameters can be expressed in the following way at first.

$$K_d = \frac{q_e}{C_e} \quad \textit{For equilibrium constant} \ (3)$$

$$\Delta G = -RT \ln K_d \quad \textit{For Gibbs energy change} \ (4)$$

$$\ln K_d = \frac{\Delta S}{R} - \frac{\Delta H}{RT} \quad \textit{For entropy and enthalpy change} \ (5)$$

where $K_c$ is the equilibrium constant, $G_o$ is the Gibbs free energy (J mol$^{-1}$), $S_o$ is the entropy (KJ mol$^{-1}$ K$^{-1}$), $H_o$ is the enthalpy (KJ mol$^{-1}$), $T$ is the absolute temperature (K), $C_o$ initial IBU concentration, $C_e$ PAC-IBU equilibrium concentration, V volume of solution, M mass of the MP-AC, and $R$ gas constant (8.314 J mol$^{-1}$ K$^{-1}$) [43].

## 3. Results

### 3.1. Characterization of PPFP

FTIR measurements are used to identify the functional groups of PPFP surface. Figure 2 shows a broad band corresponding to the OH stretching in the carboxylic group at 4327 cm$^{-1}$ and the aliphatic C-H stretching bonds at 2904 cm$^{-1}$. The band at 1613 cm$^{-1}$ is due to the presence of C=O bonds in a carboxylic acid. The peak appeared at 1310 cm$^{-1}$ due to C-O stretching in esters, ether, phenol, or carboxylic groups. The band at 1018 cm$^{-1}$ indicates the C-O-C bending [30].

Table 1 shows the B.E.T. surface area and particle size of PPFP. According to the table, the surface area of powdered prickly pear fruit peels is very tiny. This result indicated that the morphology of this biomaterial has holes. The SEM images in Figure 3 depict the general appearance of the PPFP surface, which has a high hardness and an agglomeration of small particles. They also indicate the existence of pores on a heterogeneous surface.

The identification of four crystalline phases was made possible by measuring the X-ray diffraction pattern of a dry solid sample of PPFP. These crystal phases corresponded with the diffraction patterns registered at 21.70°, as shown in Figure 4 and Tables 1 and 2. For β-alanine protein, the amino acids β-Lglutaminic acid, and α-glutamic acid, and at 15.12° and 30.30° the hydrated calcium oxalate in its syn form [47].

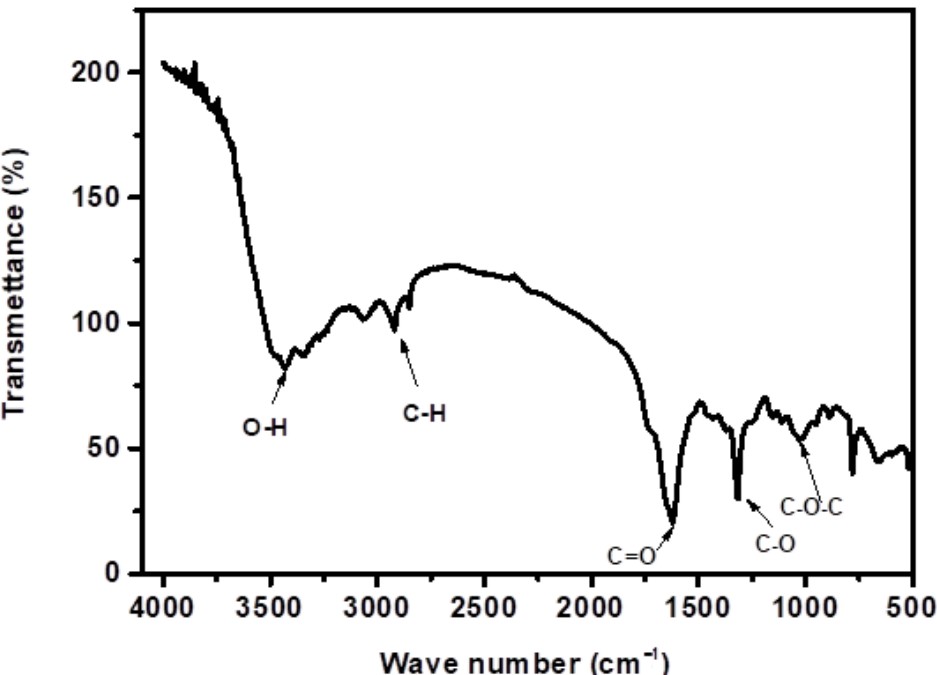

**Figure 2.** FTIR spectrum of prickly pear fruits peels (PPFP).

**Table 1.** Surface area and particle size of dried prickly pear fruits peels (PPFP).

| | |
|---|---|
| Pore size (Å) | 399.608 |
| Surface area ($cm^2$/g) | 2.54 |

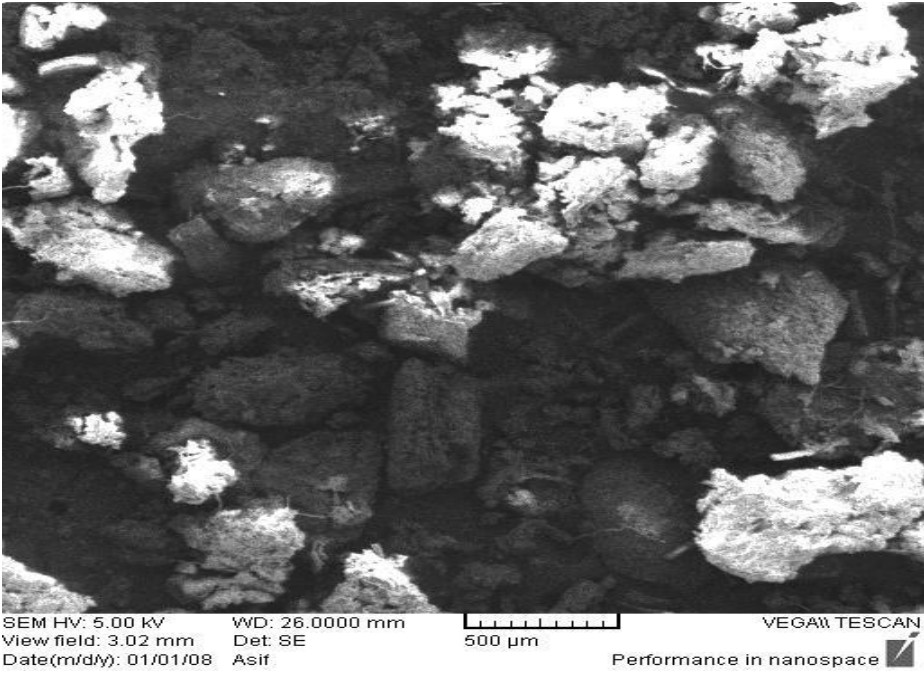

**Figure 3.** SEM image of prickly pear fruits peels surface.

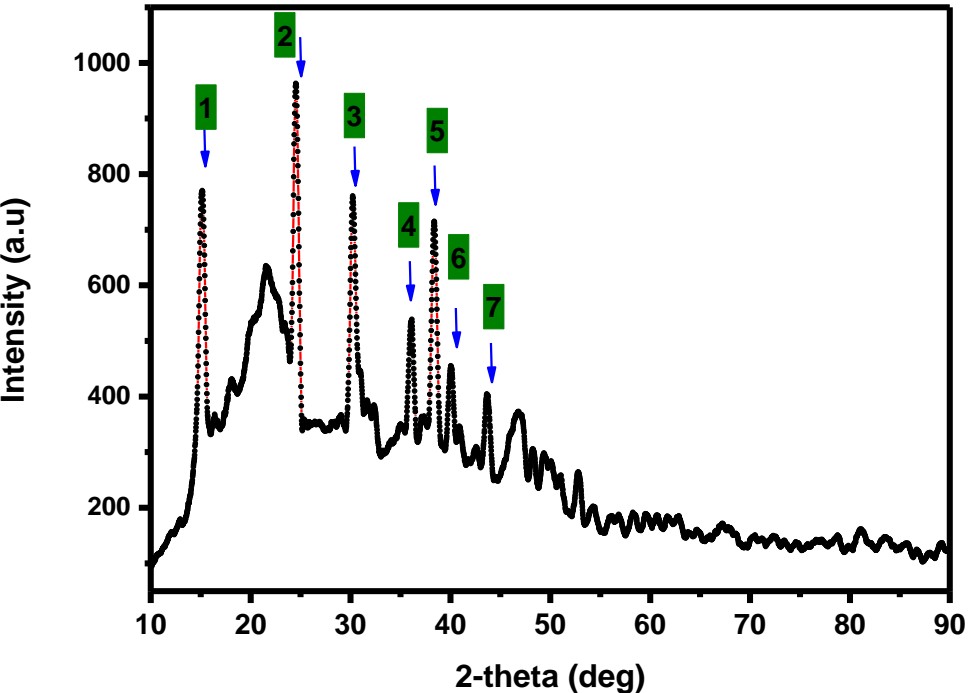

**Figure 4.** XRD of prickly pear fruits peels (PPFP).

**Table 2.** Peak position (2 theta), crystallite size, and particle size D of prickly pear fruits peels.

| Cod | Peak Position (2 Theta) | FWHM | Crystallite Size D (nm) | D nm (Average) | Particle Size (A°) |
|-----|------------------------|------|------------------------|----------------|--------------------|
| 1 | 15.1296 | 0.8198 | 9.7746 | | |
| 2 | 21.7078 | 8.6937 | 0.9304 | | |
| 3 | 30.3035 | 0.8405 | 9.7920 | | |
| 4 | 38.3851 | 0.5723 | 14.6982 | 9.9260 | 99.2609 |
| 5 | 36.0789 | 0.5554 | 15.0410 | | |
| 6 | 40.0690 | 28.1279 | 0.3006 | | |
| 7 | 40.0690 | 0.4463 | 18.9456 | | |

*3.2. Adsorption Studies*

3.2.1. The Impact of Contact Time

The removal percentage value increases with time, as seen in Figure 5, initially swiftly, later gradually, and eventually steadily. This can result from the first adsorption stage's availability of free surface sites for adsorption [31]. Due to the repelling interactions between the dye molecules in the solution and those that have been adsorbed onto the adsorbent, the remaining vacant surface sites are challenging to exploit in later phases. The equilibrium of PTZIDM dye adsorption on PPFP was attained after 60 min. The ideal contact time is 60 min, with an adsorption efficiency of 95%. The experimental findings show that the adsorption capacity of PTZIDM remains the same after 60 min; this might be because there are not many unoccupied sites available. Thus, 95% is the greatest proportion of PTZIDM elimination at symmetry time.

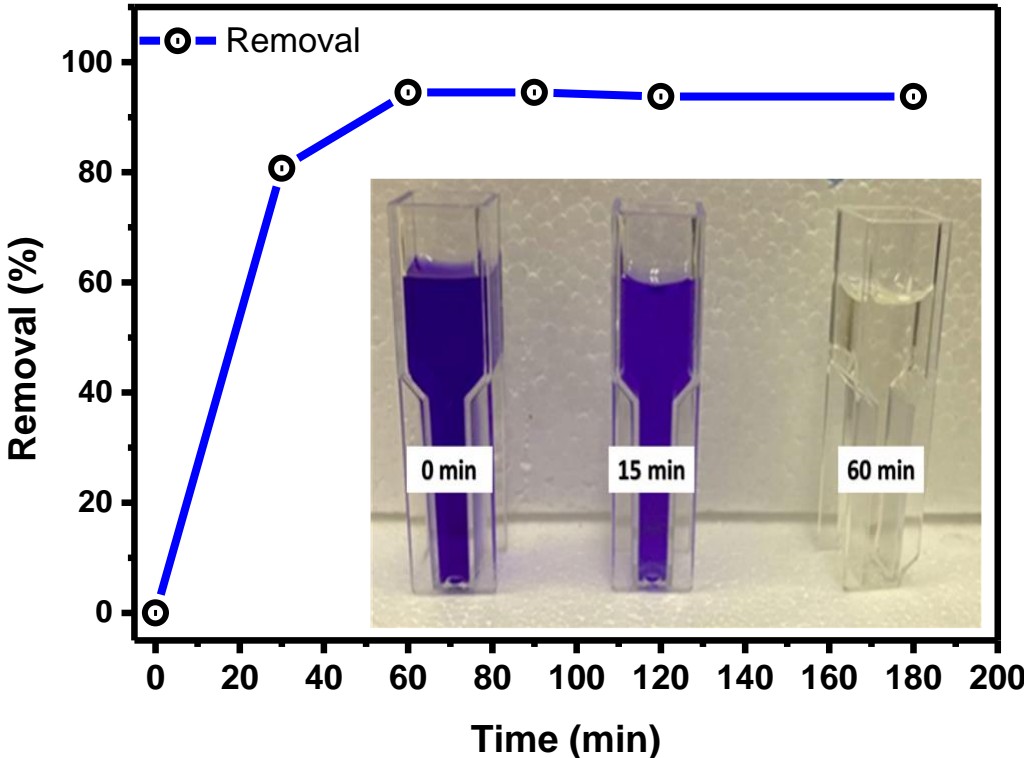

**Figure 5.** Time-dependent adsorption of the PTZIDM by the prickly pear fruits peels.

### 3.2.2. The Impact of the Initial Dye Concentration

According to Figure 6, the adsorption is obviously affected by the original dye dilution. With the original dye dilution from $1 \times 10^{-4}$ M to $2 \times 10^{-5}$ during the examination time equivalent to 60 min for PTZIDM and at room temperature, the percentage of dye adsorption declines with rise concentration. This could be the result of repelling communications between the adsorbent's surface and dye's molecule. All dye concentrations existed in the adsorption positions can interrelate with the cover of PPFP since the proportion and the percentage of accessible binding locations to the original dye is greater at lower concentrations [48]. The adsorption site is saturated at greater initial dye concentrations, preventing dye from migrating to the open cover locations on the surface, hence the adsorbent surface has a relatively lower ratio [48].

### 3.2.3. The Impact of the Adsorbent Dose

Its percentage of color removal increases as the adsorbent dose rises, as depicted in Figure 7 below. Due to increased cover area, binding sites, and pore volume, the primary cause of this behavior might be associated with the increased availability of active locations for dye–surface interactions [49]. The PPFP surface aggregation area was accessible for dye adsorption, which lengthens the diffusion channel, which may be another factor [50]. A measure of 0.1 g is the recommended dose for removing 96% of the PTZIDM dye.

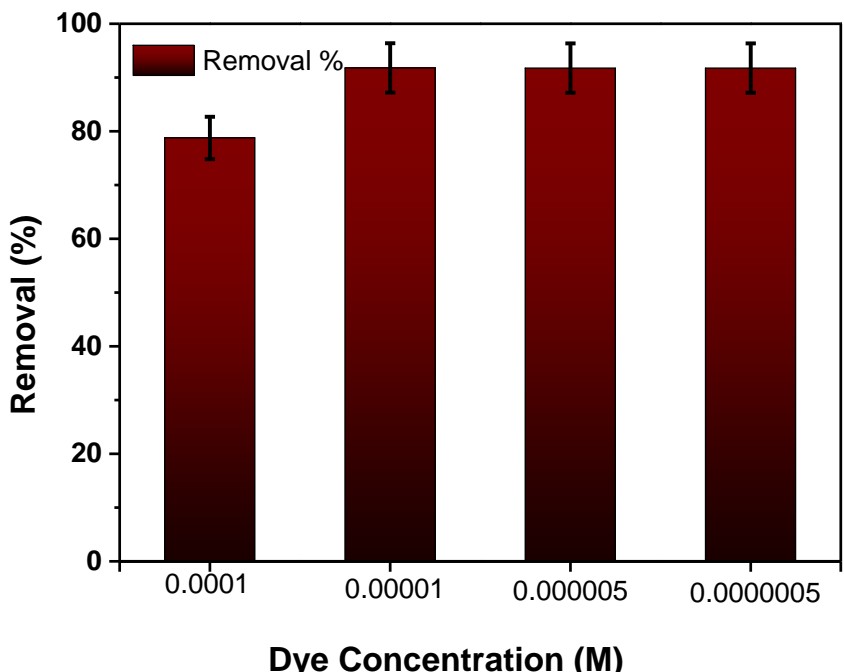

**Figure 6.** Effect of initial dye concentration at 25 °C.

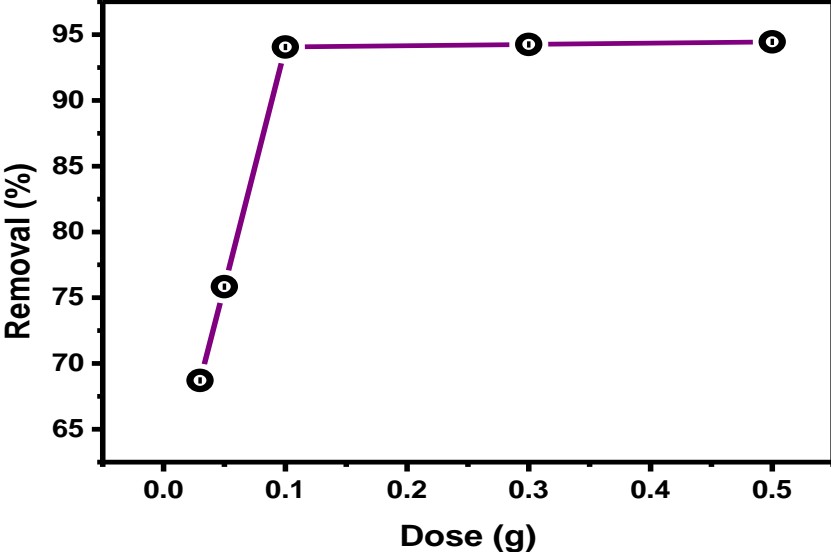

**Figure 7.** Effect of adsorbent dose at 25 °C.

### 3.2.4. The Impact of the Solution Temperature

According to Figure 8 below, the proportion of dye removal decreases as the mixture's temperature rises. This result suggests that the exothermic process is evident in the dye adsorption process, which predicts that the primary adsorption mechanism of PTZIDM onto PPFP is a chemisorption process when there are weak appealing interactions between the surface of PPFP and PTZIDM molecules [51]. For PTZIDM adsorption, the 96% dye removal occurs at 25 °C. Therefore, the ideal temperature for the adsorption procedure utilizing the produced material is determined to be 25 °C. The degree to which dye molecules diffuse from the adsorbent surface solution bulk over the external border cover and via the adsorbent's interior pores is often discouraged by a rise in solution temperature. Additionally, ion exchange rates are accelerated, and the dye ion attraction to the PPFP surface is made easier by activating the PPFP adsorbent material at decreased temperatures [51]. While the boundary layer's surface area around the adsorbent increases with temperature,

the mobility of dye molecules decreases as a consequence of the development of new active sites, which action a drop in adsorbate amount allocation resistance at the border level.

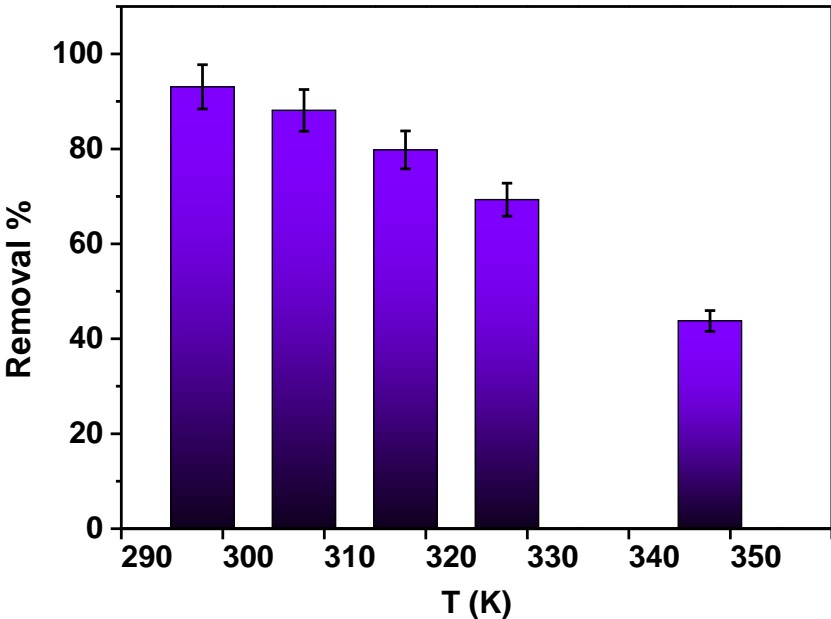

**Figure 8.** Temperature-dependent adsorption capacity of the prickly pear fruits peels for PTZIDM at pH 3.

### 3.2.5. The Impact of pH

One of the most important variables influencing adsorption capacity is the dye solution's initial pH value since it affects the surface binding sites, the dye's surface charge and the degree of ionization [52]. The dye ion absorption may be influenced by both the chemistry of the PTZIDM solution and the dynamic sites. PPFP, as an adsorbent, may have a significant amount of active locations. Over a pH from 2 to 12, the impact of solution pH values on dye extraction is examined. The removal rate of PTZIDM increased as the solution pH decreased, with 3 being the ideal pH for PTZIDM adsorption (Figure 9). The platform for better understanding and explanation of the phenomenon can be seen from the fact that donor–acceptor dye turns with a negative charge as it dissolves in water. Therefore, the positively electrified area of prickly pear increases PTZIDM adsorption in acidic conditions [52]. It is also evident that the adsorption of PTZIDM increased when the pH of the solution decreased, which is due to a raised electrostatic magnetism among the negative charge PTZIDM particles and the bioadsorbent surface that is positively charged. Consequently, the elimination of PTZIDM dye is reduced at high pH values. However, as the pH level rises, the charge on the surface shifts to the negative side, decreasing the elimination of PTZIDM. Furthermore, the great proportion elimination of the dye solution at an acidic pH may be explained by interface electrodynamics among the negatively charged PTZIDM dye anions and positively charged adsorbent [52].

In general, the surface of the adsorbent acquires negative or positive charges upon immersion in an aqueous alkaline or acidic medium (Figure 10). As a result, the negatively charged groups present in the dye molecules may be drawn to the adsorbent surface at lower pH, enhancing adsorption capacity, which decreases at higher pH. Because of this, dye removal is less effective at high pH and more effective at low pH [53].

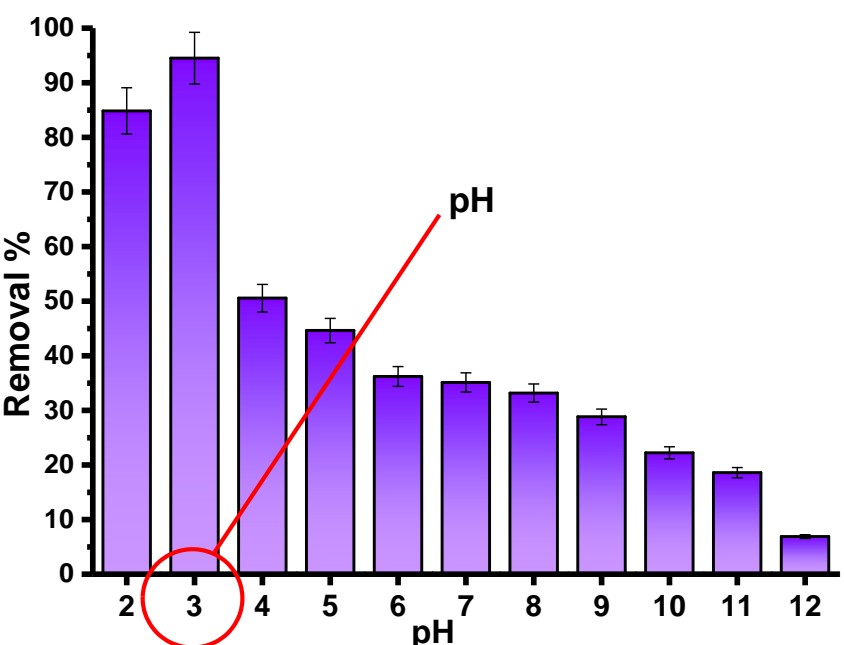

**Figure 9.** The pH-dependent adsorption of PTZIDM by the prickly pear fruit peels.

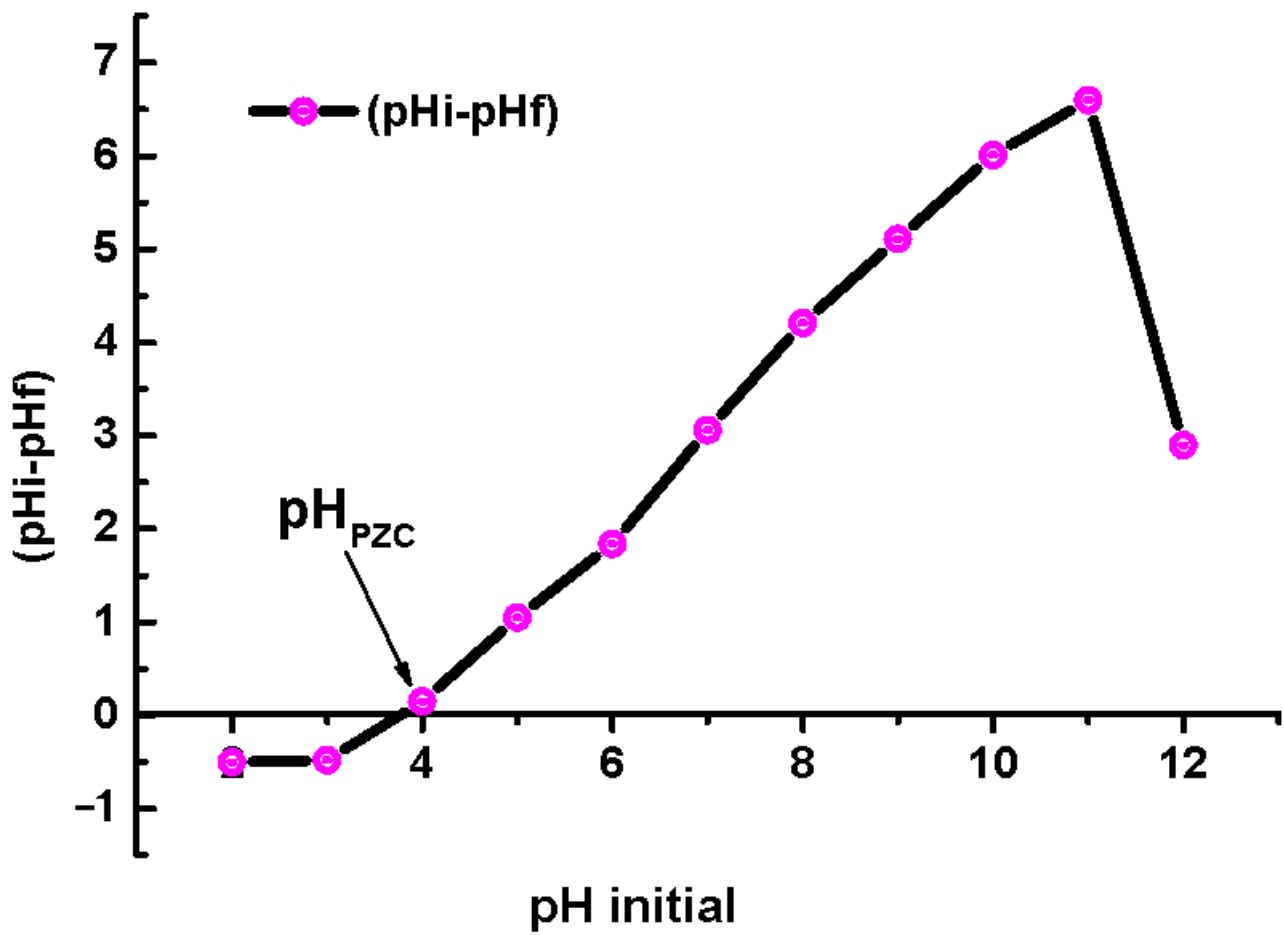

**Figure 10.** The zeta potential of prickly pear fruits peels in NaCl solution.

### 3.3. Adsorption Mechanism and Kinetic Models

The results of applying the two linearized pseudo first and second kinetics models were plotted in Figure 11. The results better fit the pseudo-second-order model than the

pseudo-first-order one. The correlation coefficients ($R^2$) of the pseudo-first-order model ($R^2$ = 0.7954) and pseudo-second model ($R^2$ = 0.9999) confirm the suitability of kinetic data with the pseudo-second-order model, in agreement with many of the previous reports in the adsorption of water pollutants [54].

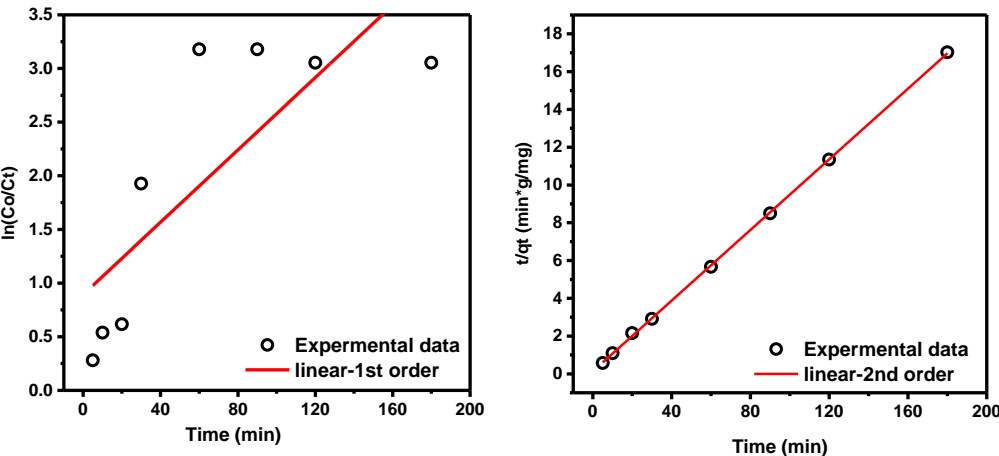

**Figure 11.** Kinetics models, pseudo-first-order (**left panel**), pseudo-second-order (**right panel**).

Moreover, all experimental parameters were calculated and listed in Table 3. The presented values exhibited good matching between the practical value ($q_e$ = 10.589 mg/g) with the theoretical value ($q_e$ = 10.767 mg/g). These values of maximum adsorption capacity correspond to the pseudo second order model, unlike its computed value of pseudo-first-order one ($q_e$ = 1.349 mg/g) [55].

**Table 3.** Kinetics parameters for pseudo-first-order, pseudo-second-order, Elovich kinetic, and intra-particle diffusion models.

| Models | Parameters | Results |
|---|---|---|
| Models | $q_e$ Exp. (mg/g) | 10.589 |
| Pseudo-first-order | $q_e$ (mg/g) | 1.349 |
| | $K_1$ (min$^{-1}$) | 0.066 |
| | $R^2$ | 0.565 |
| Pseudo-second-order | $q_e$ (mg/g) | 10.7665 |
| | $K_2$ (g/mg min) | 0.053 |
| | $R^2$ | 0.99969 |
| Elovich model | $\alpha$ (mg/g min) | 1.501 |
| | $\beta$ (g/mg) | 93,249.651 |
| | $R^2$ | 0.895 |
| Intra-particle diffusion | $K_{diff}$ (mg/g min) | 3.577 |
| | $C$ (mg/g) | 29.108 |
| | $R^2$ | 0.808 |

The graph plotted by $q_t$ against ln$t$ in the Elovich model in Figure 12 (left panel). ln$t$ in the Elovich model gave an index in terms of adsorption nature on the heterogeneous biosorbent surface, whereas the value of $R^2$ was low (0.895), indicating the property of the physisorption mechanism.

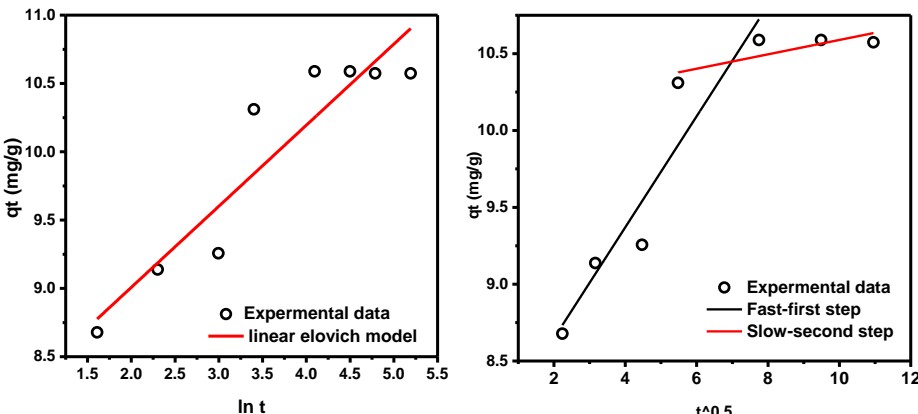

**Figure 12.** Linear Elovich model (**left panel**), intra-particle diffusion model (**right panel**).

On the other hand, the intra-partial diffusion model showed clear fitting with the experimental results of pseudo-second-order kinetics in Figure 12. The experimental data displayed two steps of the adsorption operation. The first was speed, followed by a slow step indicating that the adsorption is more than one step.

### 3.4. Adsorption Isotherms

The adsorption mechanism at the equilibrium of the organic dye was determined by combining 100 mg of dye with 50 mL of various concentrations of $1 \times 10^{-4}$ to $5 \times 10^{-7}$ M for 60 min on the digital shaker. The produced data were incorporated into the following four models: Langmuir, Freundlich, Dubinin–Raduskevich, and Temkin, as shown in Figure 13 and listed in Table S1.

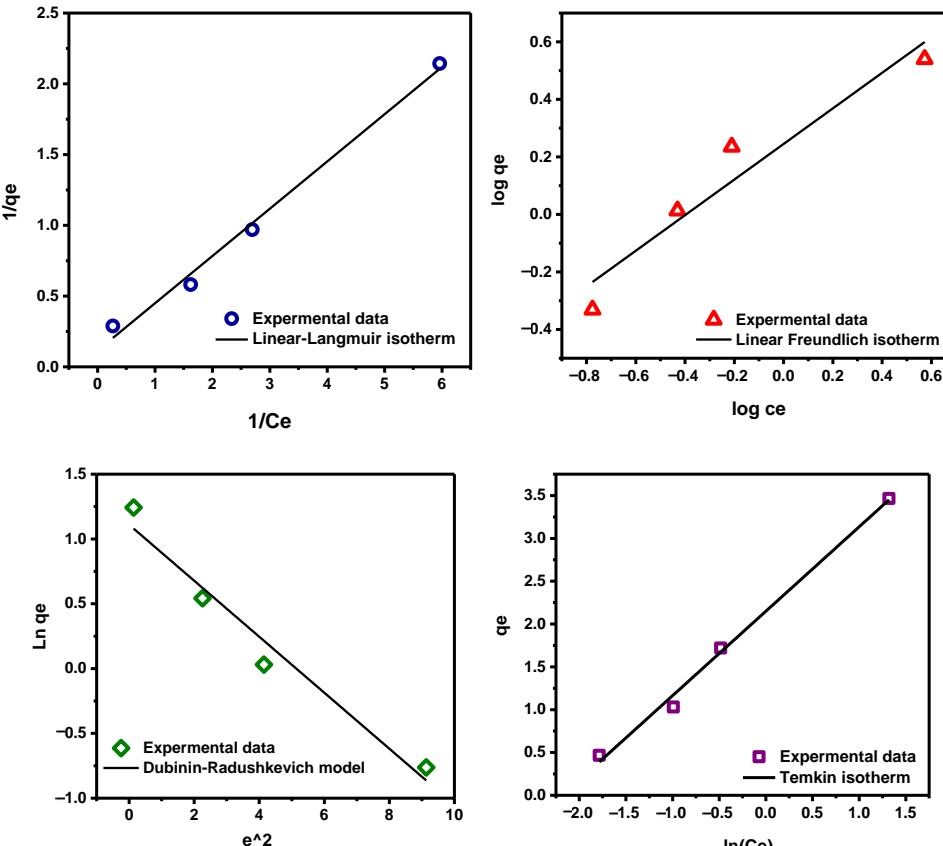

**Figure 13.** Linear-isotherm models of dye adsorption. Langmuir, Freundlich, Dubinin–Radushkevich (D-R), andTemkin isotherm.

Based on the comped results in Table S1, the $R_L$ = 0.790, which is between $R_L$ = 0 and $R_L$ = 1, meaning monolayer adsorption with uniform energetically of unlimited active sites onto the adsorbent surface. Moreover, the coefficient of correlation was equivalent to (0.992), the experimental outcome of maximum capacity for monolayer coverage was 6.69 mg/g, and the constant of Langmuir isotherm (adsorption energy) was (0.565 L/mg) to prove that adsorption isotherm was fitting to Langmuir model.

According to experimental data of applying the Freundlich isotherm model in Figure 13b, Table 4, the correlation coefficient value (0.929) indicated the unfavorable adsorption with this linear model, while the value of adsorption strength (1/n = 0.619) refers to the normal adsorption. From (n) value, the nature of the adsorption process can be determined. The value obtained was higher than 1 and lower than 10 (1.616), proving that adsorption is the favorite operation.

**Table 4.** The isotherm parameters for Langmuir, Freundlich, DB-R, and Temkin models.

| Langmuir Isotherm | | | | Freundlich Isotherm | | | |
|---|---|---|---|---|---|---|---|
| $Q_{max}$ (mg/g) | $K_L$ (L/mg) | RL | $R^2$ | 1/n | n | k f (mg/g) | $R^2$ |
| 5.301 | 0.565 | 0.790 | 0.992 | 0.619 | 1.616 | 0.414 | 0.929 |
| Dubinin–Radushkevich Isotherm | | | | Temkin Isotherm | | | |
| $q_s$ (mg/g) | $K_{ad}$ (mol$^2$/kJ$^2$) | E (KJ/mol) | $R^2$ | $A_T$ (L/m g) | $b_T$ | B (Kcal/mol) | $R^2$ |
| 162.963 | 4.464 | 0.335 | 0.964 | 8.851 | 2513.076 | 0.236 | 0.994 |

Furthermore, either one of the most isotherm models applied to determine the adsorption mechanism by the value of free energy per adsorbed molecule, physio- or chemisorption, is the Dubinin–Radushkevich model. From plotting the ($\log q_e$) Vs $\varepsilon^2$ in Figure 13c, Table 4, the energy required to remove a molecule from solution to infinity was 0.335 KJ/mol lower than 8 KJ/mol with $R^2$ = 0.694, which means that the mechanism of adsorption follows physiosorption mechanism. Additionally, the theoretical adsorption capacity ($q_s$ = 162.963 mg/g) was computed.

From plotting the Temkin isotherm in Figure 13d, the estimated results were $A_T$ = 8.851 L/g, while the heat of adsorption was B = 0.236 Kcal/mol, indicating the physical adsorption process with correlation coefficient $R^2$ = 0.994, which is higher than the correlation coefficient of Dubinin–Radushkevich model.

In conclusion, of all reports of the various factors the four models determine, the value of RL was 0 < RL < 1, and the n was 1 < n < 10, meaning the nature of adsorption is favorable. Comparing the $R^2$ of Freundlich and Langmuir isotherm suggests the adsorption is a monolayer. The highest monolayer capacity for organic dye uptake onto PPFP adsorbent was determined to be 162.963 mg/g, with a higher percentage of adsorbed organic dye molecules recorded during rapid adsorption at $1 \times 10^{-5}$ M. The adsorption energy (E) acquired from the Dubinin–Radushkevich (D-R) isotherm was 0.335 kJ/mol, indicating that the organic dye was absorbed onto PPFP via physisorption. The highest monolayer adsorption capacity ($q$max) obtained for dye uptake onto PPFP was compared to $q$max for other adsorbents cited in the literature. PPFP appears to perform better than other adsorbents previously reported.

### 3.5. The Thermodynamics of Adsorption

The results of thermodynamic parameters, which are determined to adsorb organic dye onto PPFP adsorbent, are listed in Table 5. Enthalpy ($\Delta H°$) for the adsorbate–adsorbent system was negative, suggesting that the adsorption process was exothermic. $\Delta S°$ was also negative, indicating that the random nature at the solid–liquid interface reduced during operation adsorption. The decreasing randomness is due to the association between adsorbent and adsorbate. Likewise, the process of adsorption is accompanied by a negative

value of the system's free energy ($\Delta G°$) as it becomes a spontaneous process. However, $\Delta G°$ has been increased with rising temperature as an indication of decreasing spontaneity and increasing temperature as further evidence of physical adsorption. On the other hand, the enthalpy value was usually less than 40 KJ/mol, which confirmed the physical nature of adsorption.

**Table 5.** Thermodynamics parameters for the adsorption of PTZIDM by prickly pear fruits peels at various temperatures.

| t (°C) | T (K) | ln kd | ΔG (KJ) | ΔS (KJ) | ΔH (KJ) |
|---|---|---|---|---|---|
| 25 | 298 | 1.2111 | −3.001 | | |
| 35 | 308 | 0.6174 | −1.581 | | |
| 45 | 318 | −0.0123 | 0.033 | −0.1552 | −0.1552 |
| 55 | 328 | −0.5718 | 1.559 | | |
| 75 | 348 | −1.6371 | 4.737 | | |

Adsorption Performance over Prickly Pear Fruits Peels and Other Bioadsorbents: Literature Comparison

Adsorption tests revealed that the peels of prickly pear fruits effectively eliminated dye and heavy metals from dye solutions. However, the essence of each adsorbent differs and every adsorbent has advantages and disadvantages. Data in which the adsorbent dose, pH, and highest adsorption efficiency are all taken into account were compared. The results were acquired under the most favorable experimental conditions for each work. The adsorbents listed in Table 6 were chemically activated with acids before being physically activated with high temperatures, increasing the cost of the adsorbent.

**Table 6.** Comparison of prickly pear peel adsorption capacity and other parameters obtained from the different materials reported in the literature.

| Adsorbent | Adsorption Capacity mg g$^{-1}$/Uses | Reference |
|---|---|---|
| Prickly (peel) bark of cactus fruit | Q: 150, dehydrated in direct sun 40 °C, 0.1 g, 60 min | [30] |
| Prickly pear peel (treated with $H_2SO_4$) | Q: 167 mg/g for Brilliant Green (at pH 3 and 20 °C, contact time: 240 min, $V$: 10 mL, adsorbent dose: 0.025 g) | [31] |
| Prickly pear peel activated (NaClO 12%, 323 K) | Q: 277.8 mg g$^{-1}$ for Basic blue 9<br>Q: 204.1 mg g$^{-1}$ for Basic violet 3<br>Q: 435.0 mg g$^{-1}$ for direct orange 26<br>Q: −100 mg/g for direct green 1 | [32] |
| Prickly pear peel activated (NaOH 25%, 323 K) | Q: 384.6 mg g$^{-1}$ for Basic blue 9<br>Q: 1000 mg g$^{-1}$ for Basic violet 3<br>Q: −7.75 mg g$^{-1}$ for direct orange 26<br>Q: −1.12 mg g$^{-1}$ for direct green 1 | [32] |
| Cactus fruit peel | Q: 151 acid red dye (0.1 g of BFS, concentration of 100 mg/L$^{-1}$, 2 h) | [35] |
| Prickly pear peel (Activated carbon) | Q: 294 mg g$^{-1}$ for Indigo CarmineQ: 909 mg g$^{-1}$ for Solophenyl Blue<br>Q: 416 mg g$^{-1}$ for Methylene Blue<br>Q: 312 mg g$^{-1}$ for Crystal Violet<br>20 °C, contact time | [56] |
| Prickly pear seed (Activated Carbon) | Q: 260 for methelene blue (Adsorbent dose = 0.2 g L$^{-1}$; $C_0$ = 100 mg L$^{-1}$; T = 20 °C; Ph = 7) | [57] |

**Table 6.** *Cont.*

| Adsorbent | Adsorption Capacity mg g⁻¹/Uses | Reference |
|:---:|:---:|:---:|
| (Naturel (NC) Dried Cactus (DC) | Q: 3.44 mg g$^{-1}$ for Basic blue 9<br>Q: 14.4 mg g$^{-1}$ for Basic blue 9<br>$C_o$ = 10 mg/L, Dose = 0.6 g/L for DC; (and) 2 g/L for NC, room temperature, and 60 min | [58] |
| PTZIDM | Q: 162.963 | This work |

It is significant to observe that, among the mentioned adsorbent materials, untreated prickly pear fruits peels had the third-highest sorption efficiency; moreover, the first one (activated carbon) is a material with more complex preparation methodologies; thus, greater costs are involved when compared to an acid treatment process. As a result, prickly pear fruit peels may be counted as a cost-effective adsorbent because of their simple fabrication procedure and an efficient environmental prototype for removing dye and other contaminants from wastewater.

*3.6. Regeneration Studies*

Making an adsorption process affordable and cost-effective, the adsorbent must have important properties such as the highest adsorption ability and the efficiency to regenerate and reuse various times during the adsorption process. Adsorbents with this capability are considered cost-effective and practical at the pilot scale. As a result, the utilized adsorbent was regenerated with a NaOH solution (0.1 M) and deionized water. After four cycles, the percent removal of a PPP decreased from 94.05% to 89.7%, as shown in Figure 14. There is no considerable reduction in the adsorption execution of prickly pear peel with increasing regeneration cycles because of retrieving available active sites after the renewal process. These results show that prickly pear peel could sustainably remove dye-containing wastewater.

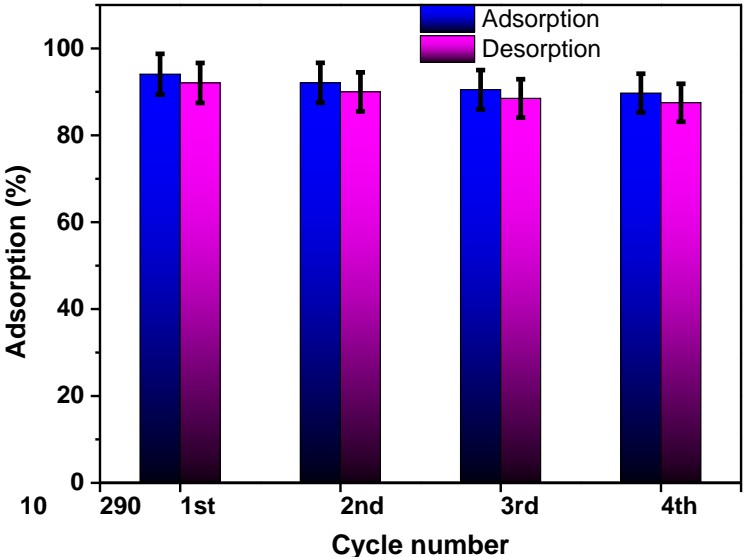

**Figure 14.** Stability of prickly pear fruits peels in adsorption of PTZIDM.

**4. Conclusions**

This study investigated the utilization of prickly pear peel as an economically feasible low-cost bioadsorbent. The results show that PPFP has a prospective adsorbent for dye elimination from aqueous emulsions because of its comparatively large surface area and many micro-pores. The effectiveness of PTZIDM's adsorption decreases with rising solution temperature and adsorbent dose while falling with rising starting dye absorption. The



quantity of dye adsorbed showed an increase with rising contact duration for all starting dye dilutions. The adsorption processes of PTZIDM also achieved equilibrium in 60 min. Acidic media was preferred for PTZIDM adsorption, with a pH of 3 being ideal. In kinetic investigations, it was discovered that the pseudo-second-order kinetic model offered reasonable suitability for the entire PTZIDM procedure of adsorption. Adsorption kinetics are often governed by several processes, with diffusion methods and tools, with intra-particle diffusion and film diffusion being the most restricting. Moreover, the Langmuir isotherm model flawlessly suits all symmetry data reported at various temperatures. The dye adsorption process is spontaneous and exothermic, and unpredictability increases near the solid or solution interface, according to negative thermodynamic parameters. The findings demonstrated that PPP is a potent and cost-effective agro-waste for eradicating PTZIDM from aqueous solutions.

**Supplementary Materials:** The following supporting information can be downloaded at: https://www.mdpi.com/article/10.3390/su15064700/s1, Table S1: Linearized form of the different kinetic and isotherm models for initial parameters estimation procedure.

**Author Contributions:** Conceptualization, F.A.M.A.-Z., B.M.A.-S., and R.M.E.-S.; methodology, F.A.M.A.-Z., B.M.A.-S., and R.M.E.-S.; software, R.M.E.-S.; validation, F.A.M.A.-Z., B.M.A.-S., and R.M.E.-S. formal analysis, F.A.M.A.-Z., B.M.A.-S., and R.M.E.-S.; investigation, F.A.M.A.-Z., B.M.A.-S., and R.M.E.-S.; resources, F.A.M.A.-Z., B.M.A.-S., and R.M.E.-S.; data curation, F.A.M.A.-Z., B.M.A.-S., and R.M.E.-S.; writing—original draft preparation, F.A.M.A.-Z. and B.M.A.-S.; writing—review and editing, F.A.M.A.-Z. and B.M.A.-S.; supervision, F.A.M.A.-Z. and B.M.A.-S.; project administration, B.M.A.-S. All authors have read and agreed to the published version of the manuscript.

**Funding:** This work was funded through Small Groups Project under grant number RGP.1/21/43.

**Institutional Review Board Statement:** Not applicable.

**Informed Consent Statement:** Not applicable.

**Data Availability Statement:** Not applicable.

**Acknowledgments:** The authors extend their appreciation to the Deanship of Scientific Research at King Khalid University Saudi Arabia for funding this work through Small Groups Project under grant number RGP.1/21/43.

**Conflicts of Interest:** The authors declare no conflict of interest.

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
