# Peer review of "Removal of Dye from Aqueous Solution Using Ectodermis of Prickly Pear Fruits-Based Bioadsorbent"

_sustainability, doi:10.3390/su15064700_

Round 1

Reviewer 1 Report (Previous Reviewer 1)

Authors address the comments properly, revisions can be accepted.

Author Response

Thanks for the acceptance and the interest in reviewing the manuscript.

Reviewer 2 Report (Previous Reviewer 2)

This is a manuscript that I revised earlier. The authors followed my suggestions, which resulted in an improvement in the quality of the work. It is necessary to additionally polish the English and technically refine the manuscript.

Author Response

Thanks for the comments. English language has been polished.

Reviewer 3 Report (Previous Reviewer 3)

In this study, the removal of dye (PTZIDM) using prickly pear fruit peels (PPFP) as an adsorbent was investigated in detail.

I would like to thank the authors for the revised manuscript.

We as reviewers are trying to help you to make your manuscript better.

I see that they have somehow tried to correct/revise the manuscript, however, I can’t see full corrections/revisions based on the comments.

Unfortunately, the manuscript (revised) still doesn't have good quality to be accepted and published in the journal of Sustainability.

They need to check the reviewers' reports and respond to the comments/corrections/questions one by one, carefully.

In order to achieve the aim of the study, the manuscript should be improved based on the discussion, and the differences according to the literature should be shown.

To understand the mechanism, the discussion must be strong.

Otherwise, the manuscript will look like a technical note rather than a research article.

Additionally, the manuscript contains some grammatical problems, and it should be checked for spelling.

Please also see the attached file where you can find my all corrections/comments, and respond to them carefully one by one.

Overall, the manuscript can only be considered an accepted paper after careful revision.

Author Response

Ms. No. Sustainability-2230885

Title of the article:

Removal of dye from aqueous solution using Ectodermis of prickly pear fruits-based bioadsorbent

RESPONSE LETTER

Comments from the Editors and Reviewers:

Reviewer #3:

Comments and Suggestions for Authors

COMMENT

RESPONSE

Q1: Provide all information for the chemicals used in this study such as company, purities etc... 

Authors accepts this comment:

NaOH, 10H-phenothiazine, 1-bromohexane, and KI are reagent grade chemicals, ≥98%, were purchased from Sigma-Aldrich (Saint Louis, MO, USA).   In addition, of 10-hexyl-10H-phenothiazine, 10-hexyl-10H-phenothiazine-3-carbaldehyde, and PTZIDM were synthesized using the previously described methods (33, 34).

Q2: Provide the information about the water such as equipment, company, model no, country, resistivity etc...

Authors accepts this comment:

Water was distilled and filtered through a

Milli-Q apparatus before use.

Q3: What crusher was used for the crushing process?

Authors accepts this comment:

“Crushed in mortar and pestle, passed through 45 micro sieves”

Q4: Provide all information for the equipment used in this study such as company, country etc...

Authors accept this comment, and the information has been added as suggestion.

“To study the chemical activities of the skin, PPFP was described using the Perkin–Elmer spectrum RXI FTIR spectrometer (Perkinelmer,

Beaconsfield, Buckinghamshire). The surface characteristics of PPFP were demonstrated using scanning electron microscope (HIROX SH

400M, Bruker, Germany). The specific total surface area was measured by the N2-sorption.

isotherms on a QuantaChrome Autosorb-6B (QuantaChrome Instruments,

Boynton Beach, FL, USA). The point of zero charge (PZC) was determined using the reported method (35). The crystalline structure was determined by an X-ray diffractometer (XRD) (Shimadzu Corporation, Kyoto, Japan).

Q5: Needs a brief explanation about the dye preparation or remove this subtitle!!!

PTZS was synthesized according to Scheme 1, as previously reported. Thus, phenothiazine was N-alkylated, followed by Vilsmeier formylation to afford the corresponding 10-hexyl-10H-phenothiazine-3-carbaldehyde in good yield. Knoevenagel condensation reaction of the carbaldehyde with 2,2’-(1H-indene-1,3(2H)-diylidene)dimalononitrile afforded the corresponding PTZS.

Q6: There is no reference about this figure in the text....

Where was it mentioned in the text?

Done

We mention the number of Scheme 1 in page 3 at line 128.

Q7: Use the same terminology throughout the manuscript...

Authors accepts this comment:

The “dosage” word in lines 99, 140, and 306 are changed to “dose” word.

Q8: Show the equations in a proper way

Authors accepts this comment:

Q9: What was the reason for calculating the removal rate?

If you calculate the adsorption capacity, it will give you everything what you need.

What do you think?

Thanks for your suggestion. The percentage of dye removal is a common and straightforward comparative index for the ability of the adsorbent to remove the dye. 

Q10: Why are the equations written in bold?

Authors accepts this comment:

The bold style has been removed.

Q11: Write the functional groups on the figure?

Authors accepts this comment, and the functional groups has been added as mentioned in figure 2.

We have also made some other changes in manuscript, e.g., formatting style, references according to your journal requirements.

We appreciate earnestly for Editors/Reviewers’ warm work and hope that the correction will meet with approval criteria.

Thank you very much again for your comments and suggestions.

Yours sincerely,

Dr. Fatimah Ali Al-zahrani

On behalf of all authors.

Reviewer 4 Report (Previous Reviewer 4)

Some corrections were made, as suggested. Therefore, I recommend its acceptance as it is.

Author Response

Thanks for the acceptance and the interest in reviewing the manuscript.

Round 2

Reviewer 3 Report (Previous Reviewer 3)

Still not good enough to be accepted...

This manuscript is a resubmission of an earlier submission. The following is a list of the peer review reports and author responses from that submission.

Round 1

Reviewer 1 Report

Fig. 3 needs to be improved significantly, scale bar missing, resolution is low, charging of specimen is too strong.

Table 2. Surface area and particle size of dried prickly pear fruits peels should be numbered as Table 1.

The insert image of Fig. 5 needs to be re-labeled, handwriting of time-interval is not acceptable.

Fig. 6 needs X axis.

The novelty of using PPFP as adsorbent needs to be discussed in detail, in the introduction.

Reversibility of adsorbent needs to be characterized, how many cycles it can be performed.

Reviewer 2 Report

1.     The title is too long

2.     It is necessary to explain the occurrence of the 10-phenothiazine based dyes in the water, its purposes, toxicity. Why were they chosen?

3.     This is a very messy manuscript, with an unacceptable amount of typos, bad sentences and wrong terms (Economic value and favourable activities(21), Fresh Prickly pear fruits is commonly consumed.; To study the chemical activities of the skin, PPFP was described using the PerkinElmer Spectrum FTIR, To investigate the surface characteristics of PPFP , its configuration was demonstrated using an HIROX SH 400 M SEM-EDS BRUKER apparatus.). Lang-muir?

4.     Although the name of the synthesized compound does not match the name of the compound in the reference by which the synthesis was done (Ref. 34), it is fascinating that when checking the validity of the synthesis by NMR and FTIR spectroscopy gave exactly the same values.

 5.     Section 2. Materials and Methods is unnecessarily burdened with theoretical details about various kinetic and isotherm models

6.     The characterization of the material is very poorly done. It is well-known that agro-waste mainly composed of cellulose, hemicellulose, lignin, pigments, protein lipids and carbohydrate. Besides, there are variety of different functional groups present in agro-waste such as hydroxyl, amino, carboxyl and sulfhydryl. Therefore, in the FTIR spectrum there are bands caused by the vibration of bonds in the molecules of cellulose, hemicellulose, lignin... and not just “presence of functional group of PPFP surface.”

7.     There is no magnification in the SEM images.

8.     It is unclear how this conclusion was made: “This result indicated that the morphology of this biomaterial has holes,”  based on:According to the table, the surface  area of powdered prickly pear fruit peels is very tiny”. Also, the analysis of the XRD peaks is not good.

9.     Table 3 shows data with an unacceptably large number of decimal figures

10.  Stating the dose as the mass of the adsorbent without the volume of the adsorbate is another imprecise data.

11.  It is not ecologically suitable to use adsorbent at pH 3 (HCl is used for adjustment), as well as regeneration with 0.1 M NaOH

Reviewer 3 Report

In this study, the authors investigated the adsorption capacity of prickly pear fruit peels (PPFP) as an adsorbent for the removal of dye (PTZIDM).

PPFP was obtained from agricultural waste.

Additionally, it was analyzed using SEM, FTIR, and BET methods.

First of all, the manuscript must be checked in terms of writing.

It needs professional support to be submitted to the journal again.

The manuscript contains some grammatical problems and needs careful reorganization in terms of grammar and spelling.

Somehow I have corrected some of them but it needs to be rechecked.

In terms of the environmental aspect, this study is very interesting and well-accepted.

Also, the authors showed well-documented experimental procedures.

Adsorption is well known in the literature, and many studies have been published about this subject for a long time.

However, they failed to present their results in the manuscript.

There need to write the manuscript based on discussion and shows the differences in results compared to the literature to achieve the scope of this study.

The discussion must be strong for understanding the mechanism.

In its current view, the manuscript looks like a technical note rather than a research paper.

I think it is better to write the manuscript again, and submit it to the journal.

Otherwise, this will make it difficult to accept it.

Meanwhile, please also see the attached file where you can find my corrections/comments, and respond to them carefully one by one.

Overall, the manuscript doesn't have good quality for being accepted in this current status.

For this reason, the paper can only be accepted after a serious correction/revision to be published in the journal of Sustainability.

Reviewer 4 Report

The present study consists in harmful dye removal from aqueous solution using Ectodermis of prickly pear fruits, an economically feasible low-cost biosorbent. This material was characterized by different techniques. The influence of relevant parameters on the adsorption process was also examined. The results obtained are quite promising taking into account the potential application.

Specific comments:

1. The manuscript should first undergo a thorough grammar correction.

2. Abstract: The authors should briefly discuss the better experimental conditions that provide maximum adsorption capacity. Thermodynamic parameters also should be described.  

3. Line 96: Opuntia ficus indica (scientific nomenclatture) in italic.

4. How is this system different to other reports to merit publication? Please, report. Several studies reported in the literature for such purpose.

5. Linearization promotes the incorrect interpretation of models due to oversimplification. The authors should perform kinetic and isotherm studies by fitting non-linear models to the experimental data to determine a variety of relevant parameters. It is more interesting to the readers. See some references that confirm such information.

https://doi.org/10.1016/j.cej.2009.09.013

https://doi.org/10.1016/j.jtice.2017.01.024

https://doi.org/10.1016/j.molliq.2019.111850

6. Random errors should be calculated for kinetic and isotherm studies as described in previous studies. These functions have also extensively decsribed in the literature (see references aforementioned).

7. Figs. 5, 6, 7, 12, and 13: Please, add error bars in all experimental results.

8. According to Fig. 13, isotherm parameters were calculated using only four experimental values. The authors should utilize a wide range of values for determining such parameters.

9. A comparative study with previous reports is highly required.